# Nutrition Meets Social Marketing: Targeting Health Promotion Campaigns to Young Adults Using the Living and Eating for Health Segments

**DOI:** 10.3390/nu13093151

**Published:** 2021-09-10

**Authors:** Clare F. Dix, Linda Brennan, Mike Reid, Tracy A. McCaffrey, Annika Molenaar, Amy Barklamb, Shinyi Chin, Helen Truby

**Affiliations:** 1School of Human Movement and Nutrition Sciences, The University of Queensland, Brisbane 4076, Australia; h.truby@uq.edu.au; 2School of Media and Communication, RMIT University, Melbourne 3004, Australia; linda.brennan@rmit.edu.au (L.B.); shinyi.chin@rmit.edu.au (S.C.); 3School of Economics, Finance and Marketing, RMIT University, Melbourne 3000, Australia; mike.reid@rmit.edu.au (M.R.); annika.molenaar@monash.edu (A.M.); amybarklamb@gmail.com (A.B.); 4Department of Dietetics and Food, Monash University, Melbourne 3168, Australia; tracy.mccaffrey@monash.edu

**Keywords:** segmentation theory, young adults, health promotion, social marketing

## Abstract

Young adults are a key target age group for lifestyle behaviour change as adoption of healthier behaviours has the potential to impact long term health. This paper arises from a multi-disciplinary research project, Communicating Health, which aims to bridge the gap between nutritionists, media, and social marketing professionals to produce the tools that may be used to improve engagement with young adults and reduce the prevalence of obesity. The aim of this paper is to provide nuanced details of the psycho-behavioral characteristics of each of these Living and Eating for Health Segments (LEHS). The design and validation of the LEHS employed a four-stage mixed methods design underpinned by the Integrated Model of Behaviour Change and incorporating sequential formative, qualitative, and quantitative phases. This paper defines the psycho-behavioural characteristics of six distinct market segments: Lifestyle Mavens, Aspirational Healthy Eaters, Balanced-all Rounders, the Health Conscious, those Contemplating Another Day, and the Blissfully Unconcerned. These psycho-behavioural characteristics are important to understand to help build our capability in designing campaigns that are specifically and purposefully targeting these different market segments of young adults. Social marketing practices can enhance the utility of nutrition and health messages to young adults in order to engage them in adopting positive lifestyle change. Tailoring health promotions to the perceived needs of sub-groups or segments of young adults should lead to increased engagement and uptake of messages and cost-efficient use of health promotion budgets.

## 1. Introduction

Overweight and obesity are increasing in young adults (YAs) and in 2014, 39% of young Australian adults are above a healthy weight [1]. This weight trajectory is underpinned by modifiable health behaviours such as poor diet and lack of physical activity [2,3]. The health risks associated with excessive fat accumulation are significant including increased risk for chronic disease [4,5,6,7], and place burden on the healthcare system [8,9,10,11]. A low awareness of obesity risk factors, a lack of effective prevention interventions and poor usage of the health system combine to make this age group challenging to engage in health promoting behaviours [12]. For many young adults, short-term tangible, often gender-specific benefits such as appearance and improvements in physical activity, were more relatable than long-term health benefits [13].

To develop effective diet and health promotion campaigns for young adults, it is important to understand the various barriers and enablers for adopting different food practices, including lifestyles, attitudes, and other psycho-behavioural characteristics that shape behaviour. This information can then be used to design campaigns and behaviour change initiatives tailored to YAs perceived needs and targeted to enhance engagement [3]. Currently, dominant health promotion practice often overlooks the need to tailor initiatives and often implements approaches that target general populations utilising a one-size-fits-all approach [14,15,16]. This one-size-fits-all approach limits campaign effectiveness, and often leaves large numbers of the audience dissatisfied, uninterested, unmotivated, and unchallenged by messages, or underexposed to messages due to poor media choices [17]. 

Market segmentation is an analytical approach commonly used within social marketing to determine sub-groups that exist within a larger group and provides a deep understanding of different types of consumers. Social marketing employs commercial marketing strategies and techniques to foster positive behaviour change in individuals to improve the health, wellbeing, and welfare of people and society. Social marketing research has identified that consumers often respond better to health initiatives and campaigns when messages and media are appropriately shaped and targeted to them based on their segment differences and characteristics [18,19]. However, despite the reported benefits of market segmentation, segmentation methods are underrepresented in many health promotion programs. Our recent systematic scoping review identified 30 studies across 27 papers in food and nutrition that used post-hoc segmentation, which does not allow for effective targeting as there is no prior understanding of the behaviour that need to change within each segment [20]. For example, of 93 interventions assessed in a recent review, a total of only 15 (16%) reported applying segmentation in social marketing programs [18].

This paper arises from a multi-disciplinary research project, Communicating Health, which aims to bridge the gap between nutritionists, media, and social marketing professionals to produce the tools that may be used to improve engagement with young adults and reduce the prevalence of obesity. The full method of this program is described in the study protocol [21]. The aim of this paper is to provide nuanced details of the psycho-behavioural characteristics of each of these Living and Eating for Health Segments (LEHS). A deeper understanding of psycho-behavioural characteristics of specific segments will enhance the development of behaviour change campaigns and intervention design. 

### 1.1. Market Segmentation

Theoretically, application of the segmentation process is related to the optimisation of limited resources so that those implementing campaigns and programs are more effective [22]. Segmentation is underpinned by the assumption that individuals that form a segment are likely to respond to a set of marketing activities (e.g., a campaign with a particular message) in a similar way, but in a way that is dissimilar to how those in other segments might respond [23]. Segmentation is driven by concerns that the limited resources available for social marketing and public health campaigns are used efficiently. It is also driven by ethical concerns related to whether people are willing to be influenced and ensuring campaigns or other programs are acceptable and meaningful to the groups they are targeted at [23].

### 1.2. Theoretical Framework

The Integrative Model of Behaviour Change (IMBC) suggests that a behaviour is more likely to occur if; (i) there is strong intention to perform the behaviour, (ii) the individual has the necessary skills and abilities to perform the behaviour, and (iii) there are no environmental constraints to completing the behaviour [24]. The IMBC model implies that different types of interventions are required dependent on which of these key components of the model are met. For example, different interventions will be required for an individual with strong intention to perform a behaviour, compared to an individual with minimal intention. In individuals with minimal intention, the model shows three key determinants of intention towards performing a behaviour: attitude, perceived norms, and self-efficacy. These three key determinants are underpinned by behavioural beliefs, normative beliefs, and efficacy beliefs. In the Communicating Health project, the behaviour of interest is healthy eating and an adapted version of the IMBC model was developed (see Figure 1). 

## 2. Materials and Methods

Ethical approval for the study has been provided by Monash University Human Research Ethics Committee (approval number 17629) and informed consent obtained by all respondents. A full description of the methods for establishing and validation of the LEHS segments have been described previously [14]. 

### 2.1. LEHS Development

In brief, the design and validation of six psycho-behavioural LEHS employed a four-stage mixed methods design [25] underpinned by the Integrated Model of Behaviour Change [24] and incorporating sequential formative, qualitative, and quantitative phases (see Figure 1).

#### 2.1.1. Literature Reviews

Three literature reviews were conducted to provide the context of the food and health environment online for young adults, this included two mixed methods systematic literature reviews investigating social media use for nutrition [26], and the impact of social media on body image and nutrition [27], and a scoping review into Indigenous Australians perspectives of the impact of social media on health [28]. 

#### 2.1.2. Qualitative Research

The literature reviews were followed by in-depth qualitative research. Data were collected from 195 respondents who completed online conversations including 20 forums, 2 challenges, 3 short polls, and ongoing journal entries to gather information about their health and well-being, especially in relation to food and the role of social media in shaping decisions about food and eating [14]. Preliminary profiles were developed using a multi-stage process with the multi-disciplinary research team and the marketing agency involved in collecting the data [14].

#### 2.1.3. Quantitative Survey

Following the formation and qualitative evaluation of the LEHS profiles, the posited segments were then subject to quantitative assessment and definition using an online survey methodology. Participants were recruited by Qualtrics^®^ in December 2018. Participant eligibility criteria included being between the ages of 18 and 24 years and currently residing in Australia. Quotas were set to achieve an Australian nationally representative sample for gender (48% female, 47% Male and 5% other) and location metro (67.3%) and regional/remote (32.7%) based on the 2016 Australian Census. Quality checks were performed on responses to screen out participants with low quality responses that were considered to not have been completed with the participants full attention. This was based on a number of factors such as an infeasibly short timeframe to have read the questions, straight lining for matrix questions, and consistency in reverse coded questions, e.g., if someone responds affirmatively to “I am eating healthily”, it is unlikely that they are not affirmative for “I am satisfied with health level in my diet”. Responses from this online survey were further used to validate the LEHS from both a methodological and an ontological perspective. Construct and nomological validity were conducted to ensure that the operationalization of the LEHS worked as a measure and to determine interrelationships with other related constructs [29].

The survey consisted of 46 closed-ended questions along with self-reported height and weight [30] (https://doi.org/10.26180/5dba10f4ec6e5) (accessed on 16 August 2021).

Questions included demographics, causes of obesity (adapted from Allison et al. [31] and Ata et al. [32], quality of life, nutrition knowledge (adapted from Mitchison et al. [33] and Flynn et al. [34]) body image satisfaction (adapted from Mitchison et al. [33]), food and food preparation skills (adapted from McGowan et al. [35]), meal skipping (adapted from Kutsuma et al. [36]) and normative beliefs, motivations (adapted from Fishbein and Ajzen [37]), and attitudes towards food and eating (adapted from Naughton et al. [38]). Participants were classified into one of the LEHS [14] by selecting which of the following written descriptions they most identified with:Lifestyle Mavens: I am passionate about healthy eating and health plays a big part in my life. I use social media to follow active lifestyle personalities or get new recipes/exercise ideas. I may even buy superfoods or follow a particular type of diet. I like to think I am super healthy.Health Conscious: I am health-conscious and being healthy and eating healthy is important to me. Although health means different things to different people, I make conscious lifestyle decisions about eating based on what I believe healthy means. I look for new recipes and healthy eating information on social media.Aspirational Healthy Eaters: I aspire to be healthy (but struggle sometimes). Healthy eating is hard work! I have tried to improve my diet, but always find things that make it difficult to stick with the changes. Sometimes I notice recipe ideas or healthy eating hacks, and if it seems easy enough, I will give it a go.Balanced All Rounders: I try and live a balanced lifestyle, and I think that all foods are okay in moderation. I should not have to feel guilty about eating a piece of cake now and again. I get all sorts of inspiration from social media like finding out about new restaurants, fun recipes, and sometimes healthy eating tips.Contemplating Another Day: I am contemplating healthy eating but it is not a priority for me right now. I know the basics about what it means to be healthy, but it does not seem relevant to me right now. I have taken a few steps to be healthier, but I am not motivated to make it a high priority because I have too many other things going on in my life.Blissfully Unconcerned: I am not bothered about healthy eating. I do not really see the point and I do not think about it. I do not really notice healthy eating tips or recipes and I do not care what I eat.

This questionnaire also collected data on respondent’s methods for communication and social media use, which will be reported elsewhere.

### 2.2. Statisticial Analysis

Individual questions were grouped together by similar characteristics for analyses (see Appendix A). Statistical analyses were conducted using IBM SPSS statistics^®^ version 25 (Armonk, NY, USA). Differences between psychographic and other lifestyle and behavioural variables assessed in the survey were evaluated using One-way ANOVA with post-hoc testing. Significance was set at *p* < 0.05 except where a Bonferroni correction was applied to pairwise comparisons.

## 3. Results

A total of 2019 young adults aged 18 to 24 years old residing in Australia completed the online survey in December 2018. Table 1 describes the characteristics of the respondents by LEHS.

### 3.1. Behavioural Beliefs

As the focus of this inquiry was related to obesity prevention, some specific questions related to what the young adults believed were the main causes of obesity. These are reported in Table 2. This demonstrated that the Balanced All-Rounders and Health Conscious were more likely to believe obesity was caused by energy imbalance than the Blissfully Unconcerned (*p* < 0.001). The Balanced All-Rounders, Aspirational Healthy Eaters, and Contemplating Another Day were more likely to believe obesity was caused by medical conditions than the Lifestyle Mavens and Blissfully Unconcerned (*p* < 0.001). The Lifestyle Mavens and Health Conscious were more likely to believe obesity is caused by lack of willpower than Contemplating Another Day (*p* < 0.001). When the causes of obesity were split by self-reported body mass index (BMI) there were significant differences between those who were overweight/obese compared to those of a healthy weight on all three stated causes of obesity. Respondents in the healthy weight group (BMI 18.5–24.9 kg/m^2^) were more likely to believe that obesity is caused by energy imbalance (*p* < 0.001) and obesity being a medically orientated condition (*p* = 0.029). Whereas those classified as overweight or obese (BMI > 25 kg/m^2^) were more likely to believe that the cause of obesity is lack of willpower (*p* = 0.013).

### 3.2. Skills in Relation to Food and Food Preparation

In Table 3, the range of skills in relation to food are shown for the LEHS. Lifestyle Mavens and Health Conscious individuals reported having better skills to plan meals ahead, use recipes, and meal-prep than other groups and the Lifestyle Mavens were the most positive about their own cooking skills which were significantly higher than the Blissfully Unconcerned and Contemplating Another Day segments (*p* < 0.05). Lifestyle Mavens and the Health Conscious segments reported being very particular about the healthiness of the food they chose to eat, and significantly different compared to Contemplating Another Day and Blissfully Unconcerned, who reported not being very particular about the healthiness of the food they eat (*p* < 0.05).

Lifestyle Mavens, the Health Conscious, and the Balanced All-Rounders also reported that they had better skills to plan grocery shopping, including using a list and shopping for specific meals, compared to Aspirational Healthy Eaters, Contemplating Another Day, and the Blissfully Unconcerned (*p* < 0.05). This pattern remained for perceived skills in budgeting, grocery shopping, including comparing prices and buying food in season compared to Aspirational Healthy Eaters, Contemplating Another Day, and the Blissfully Unconcerned (*p* < 0.05). These groups (the Contemplating Another Day and Blissfully Unconcerned) reported poorer skills in label reading and comprehension, compared to all other LEHS (*p* < 0.05). Lifestyle Mavens, the Health Conscious and the Balanced All-Rounders also thought they had high skills in resourceful cooking, including being able to prepare and cook meals with limited ingredients and time, use leftovers, and batch cook, compared to Aspirational Healthy Eaters, those Contemplating Another Day, and the Blissfully Unconcerned (*p* < 0.05).

### 3.3. Normative Beliefs and Motivation to Comply

Aspirational Healthy Eaters, those Contemplating Another Day, and the Blissfully Unconcerned were more likely to report that family, friends, and health professionals had expectations of them to eat healthier. These segments, however, were also less likely to follow advice from family, friends, and health professionals. Aspirational Healthy Eaters, those Contemplating Another Day, and the Blissfully Unconcerned also reported poorer satisfaction with their own eating habits compared to the Lifestyle Mavens, the Health Conscious, and the Balanced All-Rounders (*p* < 0.05). Perhaps not surprisingly the Blissfully Unconcerned group reported the least positive attitudes towards healthy eating. All of the LEHS reported some level of concern about the difficulty of following a healthy diet and all had similar levels of self-efficacy to do so. Nutritional knowledge was the highest in the Health Conscious and the Balanced All-Rounders (*p* < 0.05). Lifestyle Mavens reported the highest nutrition Mavenism, including being considered the nutrition “expert” in healthy eating, and people were seeking them out for information and advice (*p* < 0.05). The Blissfully Unconcerned individuals had the lowest intention to improve their diet (*p* < 0.05) and were also most likely to skip breakfast on weekends and on weekdays. Self-efficacy is an important construct of the IMBC model, Blissfully Unconcerned individuals reported lowest the levels of self-efficacy for problem solving and resilience compared to Balanced All-Rounders who reported the highest self-efficacy (*p* < 0.05) (See Table 4).

### 3.4. Environmental Constraints

The environment is also a recognized as influential in being able to choose a healthy diet. One aspect that young adults have concerns about is money, with 41.8% of respondents being in casual or part time employment, and 27.3% being unemployed. Self-reported weekly income is reported in Table 1, with 11.5% of respondents reporting having no income, with the highest percentage reported by Balanced All-Rounders (18.1%). Having insufficient money for food had been experienced by 35–53% of each segment with the Blissfully Unconcerned reporting this in over half of the group (53%). In terms of food security being a current worry, the Blissfully Unconcerned individuals reported the highest level of concern and Balanced All-Rounders reported the lowest level of concern (*p* < 0.05).

## 4. Discussion

In this article, we have provided further nuanced information on the psychographic and behavioural characteristics of previously established LEHS: Lifestyle Mavens, Health Conscious, Aspirational Healthy Eaters, Balanced All-Rounders, Contemplating Another Day, and Blissfully Unconcerned [14]. This information will enable a campaign design targeted towards young adults to be tailored to these very different segments, and therefore enhance the potential for campaign effectiveness.

To establish the LEHS, the IMBC model was used. The IMBC model focuses on a person’s intention to perform a certain behaviour, and takes into account the constraints, barriers, and capabilities (e.g., skills, environmental influences) that might prevent certain behaviours occurring. The IMBC model also recognizes the influence of factors such as personality, values, past behaviours, media exposure, demographics, and culture. These factors can all influence an individual’s beliefs, attitudes, perceived ability to act, social norms, and ultimately intentions and capacity to perform a certain behaviour. The individual factors from this model were considered in describing each LEHS and are discussed below.

Lifestyle Mavens are more likely to believe obesity is caused by a lack of willpower. They are more likely to choose healthy foods and try new foods. They perceive they have better skills in meal planning and prepping, cooking, label reading, grocery shopping and budgeting. They report being the most careful with their food choices and have the highest satisfaction levels with their eating habits, yet they believe that other people think they should eat healthier and are highly motivated to comply. True to their name, they display the most nutrition Mavenism and are most likely to report being asked for nutrition advice. The have higher perceived capability to comprehend and use food labels and be resourceful in preparing and cooking meals. As such, identifying Lifestyle Mavens could provide good role models or teachers to other groups as they motivate and therefore support others capacity to be mavens and advocates for healthy eating. Further understanding of their explicit nutrition knowledge would be needed in order to have certainty this segment would propagate evidence-based information.

Health-Conscious individuals are more likely to believe obesity is caused by disruption in energy imbalance plus individuals’ lack of willpower to resist unhealthy choices. They are more likely to choose healthy foods and try new food and they perceive they have better skills in meal planning and preparation, cooking, reading food labels, grocery shopping, and budgeting. They report being the most careful with their food choices and have high satisfaction with their eating habits and display greater nutrition knowledge; although they do not necessarily believe others think they should eat more healthily. They stated high levels of intention to eat healthier and are highly motivated to comply with extant norms surrounding healthy eating. This segment, along with Lifestyle Mavens, are highly likely to engage with health promotion campaigns due to their high levels of intention and motivations to eat healthily. An area of concern with these individuals is the potential for orthorexia and body image issues stemming from perfectionistic tendencies, as there is always room to eat healthier. A shift away from health promotion focused obesity and weight could provide a healthier approach to food and nutrition for these individuals. Supporting this segment in their current set of behaviours and helping them to accept that they are ‘enough’ could be the focus of campaigns for this group [39].

Aspirational Healthy Eaters are more likely to believe obesity is caused by medical conditions, and therefore out of individual and personal control. They are more likely to try new foods and perceive they have poorer skills in meal planning and prepping, cooking, label reading, grocery shopping and budgeting. They report satisfaction with their eating habits and were more likely to report family, friends, and health professionals with expectations of them to eat healthier. Interestingly, they were also less likely to follow advice from family, friends, and health professionals, suggesting they are less likely to be influenced by health messages. This could also be because current messages focus on individual’s necessity to enact behavioural change when change may not be within their capabilities [40]. They had higher perceived knowledge of nutrition and were likely to perceive themselves as a source of nutrition information. The challenge in targeting the Aspirational Eaters will be the potential resistance to health messaging and the perception of the individuals already being a source of nutrition information. Campaigns to this segment will need to build autonomy and self-determination and empower individuals in seeking healthy alternatives and making healthier choices when they can. The influence of poor food environments on health behaviours has been well documented [41,42], showing that obesogenic environments will lead to poorer health behaviours and ultimately poorer health outcomes. Addressing the food environment will be critical in supporting behaviour change in these Aspirational Healthy Eaters, making the healthier choice the easier choice. The medicalization and problematization of obesity [43] as a disease is likely to mitigate against this group’s behaviours, in that it will support their contention that there is nothing to be done about being overweight by an individual unless there is medical assistance.

Balanced All-Rounders are more likely to believe obesity is caused by energy imbalance or medical conditions. They are likely to try new foods and are somewhat more likely to have high self-perception of their cooking skills, meal planning, and budgeting. They are also somewhat likely to choose healthy meals, and are somewhat satisfied with their eating, but do not necessarily believe others think they should eat healthier. They have high intention to eat healthier and are somewhat motivated to comply. Balanced All-Rounders are more likely to have limited income but had the lowest level of concern regarding food insecurity and the highest self-efficacy for problem solving and resilience. Balanced All-Rounders, similar to Lifestyle Mavens, might provide a good role model or teacher to other LEHS, particularly those with lower intentions to eat healthily and lower self-efficacy. Their more balanced approach compared to Lifestyle Mavens, and higher self-efficacy could help engage Contemplating Another Day and Blissfully Unconcerned individuals. Their ability to be resilient and have high levels of self-efficacy in the face of low (no) income is something that could be examined in future research, as this would be helpful for those groups who have low levels of efficacy. Campaigns designed for this group could focus on supporting or improving existing behavioural repertoires and helping them to provide advice to others in the age cohort. Proselytizing by Mavens [44] might not be acceptable because of the psychic distance between a Maven and the Contemplating Another Day or Blissfully Unconcerned. However, some useful advice from an All-Rounder might make the healthy choice appear a little less daunting or unachievable [39].

Individuals defined by the segment of ‘Contemplating Another Day’ are more likely to believe obesity is caused by a medical condition or lack of willpower. They are less likely to try new foods and have lower self-perception of their cooking skills. They perceive that they have poorer skills in meal planning, grocery shopping, and budgeting. They have lower perceived capability to read and comprehend food labels and be resourceful when cooking. They have poorer satisfaction with their eating habits and believe others expect them to eat healthier, but their motivation to comply is lower. This group will require specific messaging and campaign action to prompt them into taking action towards healthier habits. They will also need assistance in understanding and accepting that they have agency in making healthier choices [43]. Motivating the Contemplating Another Day’s to engage in healthful behaviour will require social marketing campaigns, as well as specific interventions designed to intercede and disrupt existing behavioural repertoires. Campaigns that develop skills such as cooking, planning, and shopping are likely to be effective as these may produce desirable outcomes that are not obesity related but may lead to reductions in obesity over time. This segment may be encouraged to behave in healthful ways regardless of their beliefs if the alternative behaviour (e.g., being a better cook, saving money or time) is more desirable.

Blissfully Unconcerned individuals are likely to have the poorest skills in meal planning, grocery shopping, budgeting, cooking, and label reading. They are less likely to choose healthy foods, and report family, friends and health professionals having expectations of them to eat healthier but are less likely to follow their advice. They are most likely to have no income and more likely to report concern about and have experienced instances of food insecurity. They also report the lowest levels of self-efficacy for problem solving and resilience. Health campaigns that wish to target Contemplating Another Day or Blissfully Unconcerned individuals might need to focus on improving self-efficacy and motivation rather than specific health concerns or behaviours and provide active support to improve skills such as cooking and budgeting. Campaigns designed for this segment must avoid blaming or shaming as these types of campaign are likely to lead to reactance and avoidance, not active or rational problem solving [45].

Research has identified that consumers often respond better to health initiatives and campaigns when messages and media are appropriately shaped and targeted to them based on their segment differences and characteristics [18,19]. There is a need to adopt more sophisticated strategies for market segmentation in health promotion and move beyond using demographics or variables such as BMI as a primary grouping approach. Utilising psychographic [46], as well as behavioural segmentation, which is the grouping of people based on their behaviours such as buying or using products [15], offers far greater opportunity to engage individuals in positive behaviour change and modification [20]. Groups or clusters of young adults who, whilst they may share demographic similarities, may have significantly different lifestyles, attitudes, motivations for change and dietary behaviours and respond to different messages and creative elements in different media. Our research supports this, showing that targeting for example by gender or ethnicity only may not increase engagement, as clearly all segments are visible within all demographics. Our research demonstrates that each segment has its own unique psycho-behavioural characteristics. These characteristics will lead to entirely different responses to campaign elements. A campaign designed for a Maven will alienate the Contemplating Another Day and the Blissfully Unconcerned. A campaigned designed for the Blissfully Unconcerned may disaffect the other segments because they are already behaving in a healthful manner.

Strengths of this research includes the large sample size and the ability to use the LEHS to target groups with appropriate content. This analysis is cross-sectional in nature and describes the psycho-behavioural factors pertinent to define different segments. The nature of cross-sectional data limits its ability to determine changes between LEHS over time that may occur with transition to adulthood, experience of significant life events or indeed how motivation to change behaviour has been impacted by the COVID-19 pandemic.

Grouping target populations by demographic or disease risk factors may fail to fully identify the lifestyle and behavioural nuances that lead to increased risky behaviours. Failing to understand the nuances in young adults’ lives results in campaigns that have a one-size-fits-all approach, design messages and creative elements that lack relevance, originality, and impact, and leave young adults dissatisfied, uninterested, or unchallenged. The use of market segmentation to develop effective health promotion campaigns have been used successfully in both improving physical activity [16,47] and reducing tobacco use [48,49]. To develop effective and engaging campaigns for young adults, it is important to understand the various barriers and enablers of a healthy diet for young adults and not consider them a homogeneous age group. Their lifestyles, attitudes, and other psycho-demographic characteristics will shape their behaviour. The detailed and nuanced information reported here can be applied in practice to design campaigns and behaviour change initiatives tailored to their different segments needs and should result in more effective and cost-effective campaigns being delivered.

## Figures and Tables

**Figure 1 nutrients-13-03151-f001:**
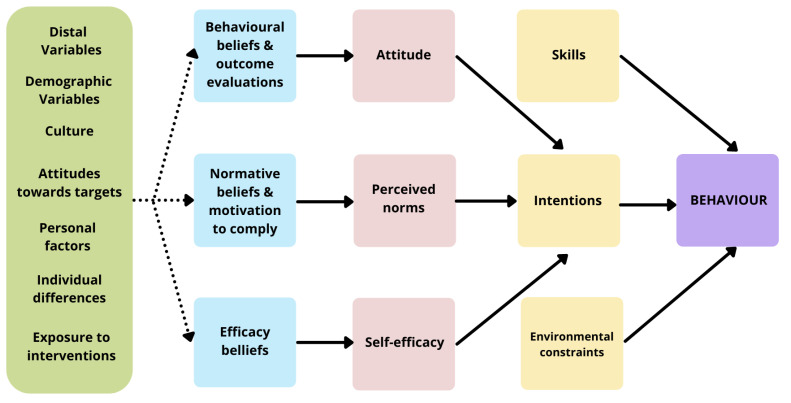
Integrated Model of Behaviour Change (IMBC), adapted from Brennan et al., 2020 [14]. Reproduced with permission.

**Table 1 nutrients-13-03151-t001:** Characteristics, demographics, and self-reported weight of Living and Eating Health Segments (LEHS) (*n* = 2019).

Characteristic	Category	Lifestyle Mavens *n* = 311 (15.4%)	Health Conscious *n* = 425 (21.1%)	Aspirational Healthy Eaters *n* = 556 (27.5%)	Balanced All-Rounders *n* = 432 (21.4%)	Contemplating Another Day *n* = 226 (11.2%)	Blissfully Unconcerned *n* = 69 (3.4%)
Age (years)		21 (2)	21 (2)	21 (2)	21 (2)	21 (2)	21 (2)
Ethnicity	Oceanian	207 (66.6%)	290 (68.2%)	433 (77.9%)	322 (74.5%)	170 (75.2%)	48 (69.6%)
	Aboriginal or Torres Strait Islander	11 (3.5%)	15 (3.5%)	17 (3.1%)	13 (3.0%)	11 (4.9%)	6 (8.7%)
	North-West European	19 (6.1%)	27 (6.4%)	42 (7.6%)	42 (9.7%)	9 (4.0%)	2 (2.9%)
	Southern and Eastern European	15 (4.8%)	22 (5.2%)	16 (2.9%)	22 (5.1%)	10 (4.4%)	1 (1.4%)
	North African and Middle Eastern	12 (3.9%)	14 (3.3%)	6 (1.1%)	10 (2.3%)	5 (2.2%)	0
	South-East Asian	7 (2.3%)	23 (5.4%)	27 (4.9%)	37 (8.6%)	14 (6.2%)	3 (4.3%)
	North-East Asian	24 (7.7%)	43 (10.1%)	25 (4.5%)	30 (6.9%)	9 (4.0%)	4 (5.8%)
	Southern and Central Asian	28 (9%)	21 (4.9%)	35 (6.3%)	23 (5.3%)	15 (6.6%)	3 (4.3%)
	Peoples of the Americas	13 (4.2%)	9 (2.1%)	11 (2%)	8 (1.9%)	4 (1.8%)	1 (1.4%)
	Sub-Saharan African	8 (3.2%)	6 (1.9%)	1 (0.2%)	4 (0.9%)	3 (1.3%)	0
	Ethnicity Not Provided	10 (3.2%)	3 (0.7%)	10 (1.8%)	6 (1.4%)	3 (1.3%)	6 (8.7%)
Education Level	Never attended school	0	3 (0.7%) ^a^	1 (0.2%) ^a^	0	0	0
	Year 8 or below	2 (0.6%) ^a^	2 (0.5%) ^a^	5 (0.9%) ^a^	3 (0.7%) ^a^	4 (1.8%) ^a^	1 (1.4%) ^a^
	Year 9 or equivalent	3 (1.0%) ^a,b^	1 (0.2%) ^a^	0	2 (0.5%) ^a^	1 (0.4%) ^a,b^	3^b^ (4.3%)
	Year 10 or equivalent	8 (2.6%) ^a,b^	12 (2.8%) ^a,b^	13 (2.3%) ^a^	19 (4.4%) ^a,b,c^	16 (7.1%) ^b,c^	9c (13.0%)
	Year 11 or equivalent	10 (3.2%) ^a^	17 (4.0%) ^a^	26 (4.7%) ^a^	23 (5.3%) ^a^	15 (6.6%) ^a^	6 (8.7%) ^a^
	Year 12 or equivalent	92 (11.3%) ^a^	164 (38.6%) ^a,b^	255 (45.9%) ^b^	191 (44.2%) ^b,c^	93 (41.2%) ^a,b^	22 (31.9%) ^a,b^
	Certificate (non-high school)	6 (1.9%) ^a,b^	5 (1.2%) ^a,b^	7 (1.3%) ^a,b^	4 (0.9%) ^a^	2 (0.9%) ^a,b^	4 (5.8%) ^b^
	Certificate I/II (non-high school)	6 (1.9%) ^a,b^	8 (1.9%) ^a^	26 (4.7%) ^a,b^	17 (3.9%) ^a,b^	9 (4.0%) ^a,b^	6 (8.7%) ^b^
	Certificate III/IV (non-high school)	29 (9.3%) ^a^	40 (9.4%) ^a^	48 (8.6%) ^a^	49 (11.3%) ^a^	27 (11.9%) ^a^	1 (1.4%) ^a^
	Advanced diploma/diploma	19 (6.1%) ^a^	34 (8.0%) ^a^	41 (7.4%) ^a^	23 (5.3%) ^a^	23 (10.2%) ^a^	3 (4.3%) ^a^
	Bachelor’s degree	75 (18.4%) ^a^	109 (26.7%) ^a^	107 (19.2%) ^a,b^	78 (18.1%) ^a,b^	29 (12.8%) ^b^	10 (14.5%) ^a,b^
	Graduate diploma/ graduate certificate	18 (5.8%) ^a^	12 (2.8%) ^a^	13 (2.3%) ^a^	8 (1.9%) ^a^	5 (2.2%) ^a^	2 (2.9%) ^a^
	Postgraduate degree	37 (11.9%) ^a^	13 (3.1%) ^b^	11^b^ (2.0%) ^b^	10 (2.3%) ^b^	2 (0.9%) ^b^	2 (2.9%) ^a,b^
	Prefer not to say	6 (1.9%) ^a^	5 (1.2%) ^a^	3 (0.5%) ^a^	5 (1.2%) ^a^	0	0
Weekly Income (AUD$)	No income	24 (7.7%) ^a^	57 (13.4%) ^a,b^	59 (10.6%) ^a,c^	40 (18.1%) ^b,c^	40 (17.7%) ^b,c^	11 (15.9%) ^a,b^
	$1–$399	89 (28.6%) ^a^	114 (26.8%) ^a^	176 (31.7%) ^a^	71 (30.3%) ^a^	71 (31.4%) ^a^	30 (43.5%) ^a^
	$400–$649	39 (12.5%) ^a^	66 (15.5%) ^a^	96 (17.3%) ^a^	28 (13.9%) ^a^	28 (12.4%) ^a^	7 (10.1%) ^a^
	$650–$999	54 (17.4%) ^a^	59 (13.9%) ^a^	90 (16.2%) ^a^	34 (15.7%) ^a^	34 (15.0%) ^a^	6 (8.7%) ^a^
	$1000–$1499	46 (14.8%) ^a^	63 (14.8%) ^a^	46 (8.3%) ^b^	23 (10.0%) ^a,b^	23 (10.2%) ^a,b^	7 (10.1%) ^a,b^
	$1500–over $3000	47 (15.1%) ^a^	45 (10.6%) ^a,b^	47 (8.5%) ^b,c^	14 (4.4%) ^b,c,d^	14 (6.2%) ^b,c,d^	3 (4.3%) ^a,b,c^
	Prefer not to say	12 (9.3%) ^a^	21 (4.9%) ^a^	42 (7.6%) ^a^	16 (7.6%) ^a^	16 (7.1%) ^a^	5 (7.2%) ^a^
Living Arrangements	One family household with only family members present	142 (45.7%)	192 (21.2%)	252 (45.3%)	192 (44.4%)	97 (42.9%)	29 (42.0%)
	Two family household with only family members present	16 (5.1%)	24 (5.6%)	30 (5.4%)	23 (5.3%)	11 (4.9%)	2 (2.9%)
	Three or more family household with only family members present	21 (6.6%)	54 (12.7%)	65 (11.7%)	51 (11.8%)	28 (12.4%)	4 (5.8%)
	One family household with non-family members present	6 (1.9%)	18 (4.2%)	20 (3.6%)	20 (4.8%)	8 (3.5%)	1 (1.4%)
	Two family household with non-family members present	9 (2.9%)	5 (1.2%)	19 (3.4%)	10 (2.3%)	7 (3.1%)	0
	Three or more family household with non-family members present	18 (5.8%)	17 (4.0%)	21 (3.8%)	14 (3.2%)	6 (2.7%)	3 (4.3%)
	Lone person household	38 (12.2%)	44 (10.4%)	39 (7.0%)	40 (9.3%)	25 (11.1%)	13 (18.8%)
	Group household	36 (11.6%)	50 (11.8%)	75 (13.5%)	62 (14.4%)	31 (13.7%)	10 (14.5%)
	Prefer not to say	25 (8.0%)	21 (4.9%)	35 (6.3%)	20 (4.6%)	13 (5.8%)	7 (10.0%)
Studying Status	Studying full-time	111 (35.7%)	151 (35.5%)	191 (34.4%)	170 (39.4%)	95 (42%)	13 (18.8%)
	Studying part-time	60 (19.3%)	86 (20.2%)	106 (19.1%)	68 (15.7%)	36 (15.9%)	13 (18.8%)
	Not studying	122 (39.2%)	173 (21.1%)	233 (41.9%)	174 (40.3%)	82 (36.3%)	36 (52.2%)
	Prefer not to say	11 (3.5%)	8 (1.9%)	11 (2.0%)	9 (2.1%)	8 (3.5%)	6 (8.7%)
Working Status	Working full-time	144 (46.3%)	141 (33.2%)	140 (25.2%)	95 (22.0%)	39 (17.3%)	13 (18.8%)
	Working part-time	75 (24.1%)	111 (26.1%)	155 (27.9%)	97 (22.5%)	53 (23.5%)	13 (18.8%)
	Working casually	33 (10.6%)	64 (15.1%)	102 (18.3%)	92 (21.3%)	39 (17.3%)	12 (17.4%)
	Not working	50 (16.1%)	101 (23.8%)	146 (26.3%)	138 (31.9%)	90 (39.8%)	26 (37.7%)
	Prefer not to say	8 (2.6%)	5 (1.2%)	12 (2.2%)	8 (1.9%)	4 (1.8%)	5 (7.2%)
Body Mass Index (kg/m^2^)		24.58 ^a,d,e^ (5.93)	23.40 (4.86) ^a^	26.04 (6.66) ^c^	23.73 (4.94) ^a,b^	25.39 (6.32) ^c,d^	26.27 (7.34) ^b,c,e^
BMI Categories	Underweight (BMI < 18.5)	28 (9.0%)	42 (9.9%)	37 (6.7%)	41 (9.5%)	16 (7.1%)	9 (13.0%)
	Healthy weight (BMI 18.5–24.9)	171 (55.0%)	275 (64.7%)	260 (46.8%)	254 (58.8%)	111 (49.1%)	30 (43.5%)
	Overweight (BMI 25.0–29.9)	72 (23.2%)	76 (17.9%)	145 (26.1%)	87 (20.1%)	53 (23.5%)	13 (18.8%)
	Obese (BMI >30.0)	40 (12.9%)	32 (7.5%)	114 (20.5%)	50 (11.6%)	46 (20.4%)	17 (24.6%)

Values in the same row and sub-table not sharing the same subscript are significantly different at *p* < 0.05 in the two-sided test of equality for column means. Cells with no subscript are not included in the test. Tests assume equal variances. Tests are adjusted for all pairwise comparisons within a row of each innermost sub-table using the Bonferroni correction.

**Table 2 nutrients-13-03151-t002:** Perceived causes of obesity by Living and Eating Health Segments (LEHS) (*n* = 2019).

Characteristic	Maximum Score	Lifestyle Mavens *n* = 311 (15.4%)	Health Conscious *n* = 425 (21.1%)	Aspirational Healthy Eaters *n* = 556 (27.5%)	Balanced All-Rounders *n* = 432 (21.4%)	Contemplating Another Day *n* = 226 (11.2%)	Blissfully Unconcerned *n* = 69 (3.4%)
Cause of obesity—Energy imbalance	10	7.0 (2.06) ^a,c^	7.3 (1.87) ^a,b^	7.2 (1.84) ^a,b,d^	7.5 (1.86) ^b^	7.2 (1.87) ^a,b,d^	6.4 (2.12) ^c,d^
Cause of obesity—Medical	10	7.4 (1.84) ^a,b^	7.8 (3.06) ^a,c^	7.9 (1.49) ^c,d^	8.2 (1.51) ^d^	7.9 (1.59) ^c,d^	6.9 (2.09) ^b^
Cause of obesity—Willpower	5	2.8 (1.18) ^a^	3.1 (1.12) ^a,b^	3.3 (3.25) ^b,d^	3.3 (1.06) ^b,c,d^	3.4 (1.05) ^d^	3.2 (1.16) ^a,b,d^

Values in the same row and sub-table not sharing the same subscript are significantly different at *p* < 0.05 in the two-sided test of equality for column means.

**Table 3 nutrients-13-03151-t003:** Skills in relation to food and food preparation of Living and Eating Health Segments (LEHS) (*n* = 2019).

Characteristic	Maximum Score	Lifestyle Mavens *n* = 311 (15.4%)	Health Conscious *n* = 425 (21.1%)	Aspirational Healthy Eaters *n* = 556 (27.5%)	Balanced All-Rounders *n* = 432 (21.4%)	Contemplating Another Day *n* = 226 (11.2%)	Blissfully Unconcerned *n* = 69 (3.4%)
Self-perception of cooking	35	23.5 (5.1) ^a^	23.6 (5.73) ^a^	22.3 (5.98) ^b,d^	22.7 (6.17) ^a,b^	20.4 (6.30) ^c^	19.9 (6.37) ^c,d^
Choosing healthy foods	5	4 (1) ^a^	4 (1) ^a^	3 (1) ^b^	3 (1) ^b^	2 (1) ^c^	2 (1) ^c^
Meal planning and prepping food	15	10.4 (2.48) ^a,c^	10.6 (2.25) ^a^	9.7 (2.41) ^b,e^	10.0 (2.64) ^b,c^	8.7 (2.12) ^d^	8.8 (2.53) ^d,e^
Shopping	15	10.4 (2.55) ^a^	10.7 (2.66) ^a^	9.8 (2.81) ^b^	10.4 (3.06) ^a^	9.5 (2.82) ^b^	9.0 (2.53) ^c^
Budgeting	20	14.0 (3.12) ^a^	14.1 (3.13) ^a^	13.0 (3.64) ^b^	13.8 (3.86) ^a^	12.4 (3.86) ^b,c^	11.2 (3.59) ^c^
Label comprehension and use	20	14.2 (3.08) ^a^	14.9 (3.19) ^a^	13.5 (3.17) ^b^	14.2 (3.39) ^a^	12.5 (3.30) ^c^	11.38 (2.93) ^c^
Resourcefulness	25	17.7 (3.67) ^a^	18.1 (3.81) ^a^	16.6 (4.09) ^b^	17.7 (4.02) ^a^	15.6 (4.30) ^b,c^	14.3 (4.0) ^c^

Values in the same row and sub-table not sharing the same subscript are significantly different at *p* < 0.05 in the two-sided test of equality for column means. Cells with no subscript are not included in the test. Tests assume equal variances. Tests are adjusted for all pairwise comparisons within a row of each innermost sub-table using the Bonferroni correction.

**Table 4 nutrients-13-03151-t004:** Normative beliefs and motivation towards food and eating in the Living and Eating Health Segments (LEHS) (*n* = 2019).

Characteristic	Maximum Score	Lifestyle Mavens *n* = 311 (15.4%)	Health Conscious *n* = 425 (21.1%)	Aspirational Healthy Eaters *n* = 556 (27.5%)	Balanced All-Rounders *n* = 432 (21.4%)	Contemplating Another Day *n* = 226 (11.2%)	Blissfully Unconcerned *n* = 69 (3.4%)
Subjective norms to healthy eating	15	8.4 (3.19) ^a,d^	7.6 (3.04) ^b^	9.1 (2.81) ^c^	7.8 (3.02) ^a,b^	9.2 (2.70) ^c^	9.5 (3.07) ^c,d^
Motivation to comply	15	10.3 (2.61) ^a^	10.3 (2.51) ^a^	9.7 (2.42) ^b^	9.7 (2.73) ^b^	8.9 (2.47) ^d^	8.3 (2.54) ^d^
Current satisfaction with healthy eating	15	10.2 (2.58) ^a^	10.2 (2.37) ^a^	7.5 (2.38) ^b^	9.1 (2.66) ^c^	7.4 (2.52) ^b^	7.7 (2.34) ^b^
Positive attitudes towards healthy eating	10	7.3 (2.14) ^a^	8.1 (1.82) ^b,c^	7.9 (1.60) ^b^	8.3 (1.57) ^c^	7.5 (1.61) ^a^	6.5 (1.91) ^d^
Heathy eating is difficult	5	3.0 (1.22) ^a,d^	3.3 (1.14) ^b^	2.6 (1.01) ^c^	3.1 (1.15) ^a,b^	2.6 (1.09) ^c^	2.6 (1.09) ^c,d^
Self-efficacy	15	11.3 (2.64) ^a,c^	11.7 (2.40) ^a,b^	11.4 (2.35) ^a^	12.1 (2.40) ^b^	11.4 (2.44) ^a^	10.3 (2.57) ^c^
Perceived nutrition knowledge/expertise	20	13.5 (3.52) ^a^	14.5 (3.18) ^b^	13.5 (3.32) ^a^	14.5 (3.24) ^b^	13.1 (3.52) ^a,c^	
Nutrition knowledge Mavenism	10	6.9 (1.71) ^a^	6.2 (1.76) ^b^	5.0 (2.02) ^c^	5.0 (2.07) ^c^	4.1 (1.92) ^d^	4.4 (1.89) ^c,d^
Intention towards healthy eating	15	10.5 (2.6) ^a^	10.6 (2.89) ^a^	11.1 (2.59) ^b^	10.1 (3.14) ^a^	9.1(3.02) ^c^	7.7(3.05) ^d^
Unhealthy eating habits (skipping breakfast during week)	5	2.8 (1.28) ^a^	2.9 (1.43) ^a^	3.2 (1.49) ^b,c^	3.0 (1.63) ^a,b^	3.4 (1.48) ^c^	3.5 (1.30) ^b,c^
Unhealthy eating habits (dinner before bed during week)	5	3.0 (1.2) ^a^	2.9 (1.23) ^a^	2.9 (1.35) ^a^	2.8 (1.45) ^a^	3.0 (1.38) ^a^	3.2 (1.19) ^a^
Unhealthy eating habits (skipping breakfast during weekend)	5	2.9 (1.25) ^a,b^	2.8 (1.42) ^a^	3.2 (1.50) ^b,d^	2.9 (1.64) ^a,b^	3.3 (1.50) ^b,c,d^	3.6 (1.25) ^d^
Unhealthy eating habits (dinner before bed during weekend)	5	3.2 (1.20) ^a^	2.8 (1.27) ^b,c^	2.9 (1.40) ^b^	2.6 (1.46) ^c^	2.8 (1.39) ^b,c^	3.1 (1.17) ^a,b,c^

Values in the same row and sub-table not sharing the same subscript are significantly different at *p* < 0.05 in the two-sided test of equality for column means. Cells with no subscript are not included in the test. Tests assume equal variances. Tests are adjusted for all pairwise comparisons within a row of each innermost sub-table using the Bonferroni correction.

## Data Availability

The data are not publicly available due to privacy and ethical considerations as participants did not consent for their information to be made accessible on a public repository. A copy of the survey questions are available at: https://doi.org/10.26180/5dba10f4ec6e5 (accessed on 16 August 2021).

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
