# Peer review of "Nutrition Meets Social Marketing: Targeting Health Promotion Campaigns to Young Adults Using the Living and Eating for Health Segments"

_nutrients, 2021, doi:10.3390/nu13093151_

Round 1
Reviewer 1 Report
This study provides important information for tailored interventions by market segments in the young population, and is fairly well described.
However, there are some issues that need improvement.
- Introduction: The authors describe only overweight and obesity in the introduction. However, a healthy diet in the young population is closely related not only to obesity but also to non-communicable chronic diseases such as hypertension and diabetes. Please provide additional descriptions for these related diseases.
- Discussion: The authors briefly described the ineffectiveness of the one-size-fits-all approach, but it is necessary to describe examples and inefficiencies of this approach additionally, and it is also necessary to reinforce examples of successful interventions for reader's understanding.
Author Response
Thank you for your review. Please find our responses in the attached document.

Reviewer 2 Report
Revision of the manuscript "Nutrition meets social marketing: Targeting health promotion campaigns to young adults using the Living and Eating for Health Segments"
The manuscript deals with an interesting and topical subject. The need for a change in dietary patterns among the young population.
Having read the manuscript, I suggest a number of small modifications that may help to improve the paper.
Abstract
I consider that the abstract is not adequate for an article like this. The authors should make it more scientific, i.e. the abstract should include the objectives, the sample, the materials used, the main results and the most important conclusions. In this way, the future reader can understand the subject of the paper with a quick reading.
Introduction
The section is well-founded, although the objective of the paper should be specified in more detail.
In line 87 there is an error, the point (ii) is missing.
Material and methods
Would it be possible for the authors to add the survey in an appendix? This would help to compare the results of the Australian population with other countries.
After these small contributions and future modifications I suggest the publication of the manuscript.
Author Response

(The authors gave the same response as above.)

Round 2
Reviewer 1 Report
There are additional comments.